# Development and Protective Efficacy of a Novel Nanoparticle Vaccine for Gammacoronavirus Avain Infectious Bronchitis Virus

**DOI:** 10.3390/vaccines13080802

**Published:** 2025-07-28

**Authors:** Ting Xiong, Yanfen Lyu, Hongmei Li, Ting Xu, Shuting Wu, Zekun Yang, Mengyao Jing, Fei Xu, Dingxiang Liu, Ruiai Chen

**Affiliations:** 1College of Veterinary Medicine, South China Agricultural University, Guangzhou 510640, China; bearvet@163.com (T.X.); lyf20130327wd@163.com (Y.L.); wushuting2@126.com (S.W.); xufei0918@193.com (F.X.); 2Zhaoqing Branch of Guangdong Laboratory of Lingnan Modern Agricultural Science and Technology, Zhaoqing 526238, China; lihm11@163.com (H.L.); 15662032466@163.com (T.X.); zekunyang961224@163.com (Z.Y.); 18695823208@163.com (M.J.); 3Key Laboratory of Manufacture Technology of Veterinary Bioproducts, Ministry of Agriculture and Rural Affairs, Zhaoqing 526238, China; 4Zhaoqing Dahuanong Biology Medicine Co., Ltd, Zhaoqing 526238, China; 5School of Mathematics and Physics, Hebei University of Engineering, Handan 056038, China; 6Integrative Microbiology Research Centre, South China Agricultural University, Guangzhou 510642, China

**Keywords:** IBV, nanoparticle vaccine, protective efficacy

## Abstract

**Background**: Infectious bronchitis virus (IBV) is a gammacoronavirus that causes a highly contagious disease in chickens and seriously endangers the poultry industry. The *GI-19* is a predominant lineage. However, no effective commercially available vaccines against this virus are available. **Methods**: In this present study, the CHO eukaryotic and the *E.coli* prokaryotic expression system were used to express S1-SpyTag and AP205-SpyCatcher, respectively. Subsequently, the purified S1-SpyTag and AP205-SpyCatcher were coupled to form the nanoparticles AP205-S1 (nAP205-S1) in PBS buffer at 4 °C for 48 h. S1-SpyTag and nAP205-S1 were formulated into vaccines with white oil adjuvant and employed to immunize 1-day-old SPF chickens for the comparative evaluation of their immune efficacy. **Results**: The nAP205-S1 vaccine in chickens induced robust IBV-specific humoral and cellular immune responses in vivo. Importantly, the humoral and cellular immune responses elicited by the nAP205-S1 vaccine were more robust than those induced by the IBV S1-SpyTag vaccine at both the same dose and double the dose, with a notably significant difference observed in the cellular immune response. Furthermore, experimental data revealed that chicken flocks vaccinated with nAP205-S1 achieved 100% group protection following a challenge, exhibiting a potent protective immune response and effectively inhibiting viral shedding. **Conclusions**: These results reveal the potential of developing a novel nanoparticle vaccine with broadly protective immunity against GI-19 IBV.

## 1. Introduction

Coronaviruses, which are enveloped viruses containing single-stranded positive-sense RNA (+ssRNA), are members of the order Nidovirales [1] and are categorized into four distinct genera: alpha-, beta-, gamma-, and deltacoronavirus. The infectious bronchitis virus (IBV) is acknowledged as the earliest identified representative of the Gammacoronavirus genus and was originally isolated in 1931 in North Dakota, United States [2]. The spike (S) glycoprotein is cleaved by host proteases, generating the S1 and S2 subunits. Notably, the S1 subunit contains the receptor-binding domain (RBD), which plays a critical role in recognizing and binding to specific receptors on the surface of host cells. Furthermore, the S1 subunit, particularly the RBD, serves as the primary target for neutralizing antibodies and effectively elicits protective immune responses in the host.

The high error frequency during IBV genome replication, in conjunction with the frequent occurrence of recombination events, leads to the ongoing emergence of a wide range of IBV variants [3,4,5]. The most recent classification system based on the full length of S1 further divides IBV into seven genotypes (*GI~GVII*) and thirty-three distinct lineages [6]. Currently, the dominant strains circulating in China include *GI-1*, *GI-7*, *GI-13*, *GI-19*, *GI-22*, *GI-28*, *GVI-1*, and several mutated variants. Among these, the *GI-19* genotype has emerged as the most prevalent strain, representing 61.8% of reported cases and showing a consistent upward trend in prevalence over the past five years [5]. Infectious bronchitis virus primarily affects chickens and demonstrates broad tissue tropism, leading to impairments in the respiratory, urogenital, and digestive systems [2], and some strains possess a fatality rate exceeding 60% [7]. Vaccination serves as the primary means for the prevention and control of infectious bronchitis virus (IBV). Nevertheless, due to the numerous genotypes of IBV and the poor cross-protective response among different genotype strains, the existing commercial vaccines frequently fail to confer effective immunity or are only partially efficacious [8,9]. There is an urgent necessity to develop highly efficient and novel vaccines to fulfill the requirements of clinical prevention and control.

The SpyTag-SpyCatcher system is derived from the CnaB2 domain, which is the second immunoglobulin-like collagen-binding region found in the fibronectin-binding protein of Streptococcus pyogenes (Spy). This system achieves stability through the formation of an intramolecular isopeptide bond [10]. The CnaB2 domain was rationally designed into the SpyTag peptide (comprising 13 amino acids) and the SpyCatcher protein (consisting of 116 amino acids). When the reactive pair is mixed under a broad range of physiological conditions, an irreversible amide bond is spontaneously formed between the aspartic acid residue^117^ on SpyTag and the lysine residue^31^ on SpyCatcher [11]. The SpyTag-SpyCatcher technology has extensive applications and has recently been successfully utilized in various protein-based synthetic nanoparticle scaffolds [12,13]. The 180-amino-acid SpyCatcher-AP205 particle is assembled from the phage AP205 capsid protein fused with SpyCatcher. This technology has been widely applied in various fields, including the production of vaccine nanoparticles [14], the stabilization of enzymes [15], the modular assembly of antibodies [16], and the site-specific labeling of antibodies [17].

In this study, we fabricated the fusion protein S1-SpyTag reacted with the fusion protein AP205-SpyCatcher in PBS buffer at 4 °C for 48 h to obtain the nAP205-S1. Moreover, the immunogenicity of nAP205-S1 was compared and evaluated through animal immunization experiments. We investigated the ability of the nAP205-S1 and S1-SpyTag vaccines to induce IBV-specific antibodies, a cellular immune response, and protective efficacy against IBV in SPF chickens. Our results demonstrated that the nAP205-S1 vaccine elicits higher humoral and cellular immunity than the IBV S1-SpyTag vaccine and the protection rate against the challenge reaches as high as 100%.

## 2. Materials and Methods

### 2.1. Cells, Animals, and Ethics Statements

The CHO-K1 cell line was obtained from Professor Li Zuosheng at the Zhaoqing Branch of the Guangdong Laboratory for Modern Agricultural Sciences and Technology. One-day-old specific-pathogen-free (SPF) chickens and SPF chicken embryos were sourced from Xinxing Dahuanong Poultry Egg Co., Ltd. (Yunfu, China) for use in the immunological efficacy evaluation. This research did not involve any endangered or protected animal species and was conducted under the approval of the Animal Experiments Committee of Zhaoqing Dahuanong Biopharmaceutical Co., Ltd. (Zhaoqing, China) (LL-G-20250301-01).

### 2.2. Cloning, Expression, and Purification of Recombinant S1-SpyTag and AP205-SpyCatcher Proteins

To create a recombinant IBV S1-SpyTag-SpyTag construct, the self-signal peptide constituted by amino acids 1 to 18 of the S1 protein was substituted with the mouse IgGκ signal peptide to improve the translation efficiency of the S1 protein. Meanwhile, SpyTag and 6×His tags were successively added to the C-terminus of the sequence. The IBV S1-SpyTag-SpyTag fusion protein was generated through the gene synthesis of cDNA encoding the Kozac sequence, the IgGκ signal peptide (amino acid residues 1–18), the S1 subunit (amino acid residues 19–535), SpyTag (AHIVMVDAYKPTK), and a 6×His tag, optimized with CHO codons by GenScript. The S1 subunit, SpyTag, and 6×His were linked through a linker (GGGS)2. The cDNA was cloned into pXJ40 expression vector modified by replacing the AmpR resistance with NeoR/KanR resistance at EcoRI and BglII sites. The resulting plasmid was named pXJ40(NeoR/KanR)+-S1-SpyTag (Figure 1A). CHO-K1 cells (Chinese Hamster Ovary) in the logarithmic growth phase were trypsinized and seeded into 6-well plates. The cells were cultured in a humidified incubator at 37 °C with 5% CO_2_ until approximately 80% confluence was achieved, at which point transfection was performed. Forty-eight hours post-transfection, the supernatant was harvested for analysis of transient protein expression levels. Subsequently, the supernatant was discarded and 2 mL of complete medium (Ham’s F-12K supplemented with 10% fetal bovine serum, 1% penicillin-streptomycin, and 700 μg/mL G418) was added to each well. Cell morphology and growth were monitored daily. Due to metabolic activity, the medium was replaced every two days. This process continued until cell clusters formed in the transfected groups, while the negative control group (cells without plasmid transfection) exhibited complete cell death. Single-cell colonies were subcloned by limited dilution to screen for high-expression stable cell lines. These stable cell lines were then adapted to serum-free suspension culture using a stepwise adaptation method. After 12 days of serum-free suspension culture, the cell supernatant was collected and the expression level of S1-SpyTag was evaluated by SDS-PAGE.

The capsid protein of Acinetobacter phage AP205 (Gene ID: 956335) was genetically linked to the SpyCatcher at its N-terminus to generate the nanoparticle AP205-SpyCatcher. This construct was then subcloned into the pET28a+ expression vector using the EcoRI and XhoI restriction enzyme sites located at the termini (Figure 2A). The resulting AP205-SpyCatcher plasmid was introduced into *E. coli BL21*(DE3) cells, which were initially cultured at 37 °C in LB medium supplemented with kanamycin until the OD600 reached 0.6 to 0.8. Protein expression was subsequently induced by adding 0.5 mM IPTG, followed by incubation for 19 h at 16 °C. Bacterial cells were collected by centrifugation at 4 °C and 8000 rpm for 10 min. The precipitate was washed with PBS and the inclusion bodies were dissolved in 8 M urea. Subsequently, ultrasonication was performed at 200 W with a duty cycle of 3 s on and 6 s off until the bacterial suspension became clarified. The protein supernatant was then collected by centrifugation at 12,000 rpm for 20 min and loaded on SDS-PAGE gels to examine the levels of AP205-SpyCatcher protein expression.

The supernatant containing the synthesized AP205-SpyCatcher protein was filtered through a 0.45 μm filter membrane. The filtrate was then loaded onto a Ni-NTA affinity column and incubated overnight at 4 °C with gentle inversion to ensure efficient binding of the target protein. The column was subsequently washed with a buffer containing 50 mM imidazole to remove non-specifically bound proteins, followed by elution of the target protein using a buffer containing 250 mM imidazole. Aliquots of 50 μL from both the flow-through and elution fractions were collected, mixed with SDS-PAGE loading buffer, and subjected to boiling water bath treatment. After centrifugation, the samples were analyzed by SDS-PAGE electrophoresis to evaluate the purification efficiency of the target protein.

### 2.3. Generation and Verification of nAP205-S1

To display the S1 antigen on the capsid AP205 surface, purified IBV S1-SpyTag-SpyTag and AP205-SpyCatcher were mixed at 1:2 concentration ratios with auto-assembled nAP205-S1 using the SpyTag/SpyCatcher technology. The mixture was incubated in PBS buffer at 4 °C for 48 h. The coupling products were characterized and validated using SDS-PAGE and transmission electron microscopy (TEM).

### 2.4. Vaccine Preparations

The vaccine preparation process is described as follows: First, prepare the oil phase by combining 94 parts of injection-grade white oil (Mobil, Spring, TX, USA) with 1.5 parts of aluminum stearate in an oil-phase preparation tank and heat the mixture to 80 °C. Subsequently, add 6 parts of Span-80 (Sigma-Aldrich, Alexandria, VA, USA) and maintain the temperature at 116 °C for 30 min. After cooling, store the oil phase for subsequent use. Simultaneously, prepare the water phase by introducing 4 parts of Tween-80 (Sigma-Aldrich, Alexandria, VA, USA) into a solution preparation tank, followed by sterilization and cooling for later use. Prepare three antigen solutions by diluting conjugated AP205-S1 (with an absolute S1-SpyTag concentration of 250 μg/mL), 250 μg/mL S1-SpyTag protein solution, and 500 μg/mL S1-SpyTag protein solution with sterile physiological saline at a ratio of 3:5. Introduce 96 parts of the three equally volumed antigen solutions into the solution preparation tank while continuously stirring until the Tween-80 is fully dissolved. Finally, emulsify the mixture by transferring 2 parts of the oil phase into an emulsification tank, initiating slow motorized stirring, and gradually adding 1 part of the water phase. Following addition, emulsify the mixture at 3200 revolutions per minute for 30 min.

### 2.5. Immunization of Chickens and Virus Challenges

SPF chickens (n = 6/group) were allocated to four groups within negative pressure isolators and vaccinated subcutaneously at 1 day old with 200 µL of nAP205-S1 vaccine (6 µg), S1-SpyTag vaccine (6 µg), S1-SpyTag vaccine (12 µg), and PBS Con, respectively. An immunization booster was conducted using the same approach and dose at 10 days old.

At 28 days post-immunization (dpi), six chickens in each group were challenged with ~10^5.5^ EID_50_ of CK/CH/JS/TAHY strain by the nasal–ocular route. Furthermore, two unimmunized SPF chickens were introduced into each of the nAP205-S1 and PBS immunization groups post-challenge to evaluate their virus shedding. Following inoculation, the challenged and control chickens were monitored daily for clinical signs including tracheal rales, wheezing, nasal discharge, or mortality over an 8-day period. At 8 days post-challenge, all surviving birds were humanely euthanized and immediate necropsy examinations were conducted. Kidney and tracheal tissues from the four challenged groups were harvested separately and preserved in 10% neutral buffered formalin for subsequent histopathological evaluation. The histopathological scoring criteria were defined as follows: Trachea—0 points, no damage; 1 point, mild loss of cilia in the tracheal epithelium; 2 points, inflammatory cell infiltration; and 3 points, shedding of tracheal epithelial cells. Lung—0 points, no damage; 1 point, minor bleeding in the alveolar cavity and shedding of alveolar epithelium; 2 points, extensive bleeding and inflammatory cell infiltration; and 3 points, interstitial hyperplasia with widened alveolar septa. Kidney—0 points, no damage; 1 point, hemorrhage and shedding of renal tubular epithelial cells; 2 points, inflammatory cell infiltration; and 3 points, degeneration and necrosis of renal tubules and collecting ducts.

### 2.6. Antibody Detection and Cellular Immunoassay

Serum samples were randomly collected from six chickens in each group at 7, 14, 21, and 28 days post-infection (dpi). The concentration of immunoglobulin G (IgG) specific to infectious bronchitis virus (IBV) in the serum was assessed using an indirect enzyme-linked immunosorbent assay (ELISA) (self-developed). Serum samples and goat anti-chickens IgG-HRP (Solarbio, Beijing, China) were 100-fold and 5000-fold diluted, respectively, and then detected by ELISA; positive sample was defined as an S/P (sample-negative)/(positive–negative) value > 0.24.

At 28 days post-infection, three anticoagulated peripheral blood samples were randomly collected from each group of chickens. These samples were subjected to flow cytometric analysis (Beckman Coulter, Carlsbad, CA, USA) to evaluate the levels of CD3+, CD4+, and CD8+ T cell subsets using the following monoclonal antibodies: mouse anti-chicken CD3, mouse anti-chicken CD4, and mouse anti-chicken CD8α (Southern Biotech, Birmingham, AL, USA) [18].

### 2.7. Statistical Analysis

The data were analyzed using the GraphPad software package. Statistical differences between the indicated groups and their corresponding control groups were evaluated through two-way analysis of variance (ANOVA). The levels of statistical significance are indicated by *p* values, where ns represents non-significance, * *p* < 0.05, ** *p* < 0.01, *** *p* < 0.001 and **** *p* < 0.0001.

## 3. Results

### 3.1. Expression and Purification of IBV S1-SpyTag

The 130 kDa IBV S1-SpyTag protein was synthesized using a serum-free suspension CHO stable cell line developed as follows. The recombinant expression plasmids pXJ40(NeoR/KanR)-S1-SpyTag (4 μg) were transfected into CHO-K1 cells and the transfected cells were selected by G418. Approximately 30–40% of the transfected cells remained viable on day 4 of G418 drug selection. They exhibited pronounced clonal growth with a significant increase in their cell number on day 8, whereas all cells in the control group (transfected with an empty vector) had died by then (Figure 1B). Five large single-cell colonies were individually selected and transferred to a 24-well plate for expansion culture. Upon reaching confluence, the supernatants were collected and the protein expression levels in each well were quantitatively analyzed by Western blot, demonstrating that the target band in the supernatant from well No. 4 was significantly higher than that from other wells (Figure 1C). Cells from well No. 4 were then subcloned and clone No. 12 displayed the highest protein expression level (Figure 1D). This clone was successfully adapted to a serum-free suspension culture through a stepwise adaptation process (Figure 1E,F) and stable protein expression was maintained over 15 passages (Figure 1G). Following systematic medium optimization, the Han1 medium was identified as the most suitable medium for the high-level suspension expression of IBV S1-SpyTag (Figure 1H). After 12 days of serum-free suspension culture, the supernatant was harvested, concentrated by ultrafiltration, and purified via affinity chromatography, generating the recombinant IBV S1-SpyTag protein for the subsequent vaccine preparation (Figure 1I).

### 3.2. Expression and Purification of Acinetobacter Phage AP205 Capsid Protein AP205-SpyCatcher

The 37 kDa AP205-SpyCatcher protein (Figure 2A) was synthesized and purified from bacteria. The recombinant protein expression was induced with 0.5 mM IPTG at 16 °C for 19 h. A minor proportion of the fusion protein AP205-SpyCatcher was synthesized as soluble protein (Figure 2B). The further purification of the target protein in the supernatant through affinity chromatography successfully obtained the high-purity fusion protein AP205-SpyCatcher (Figure 2C).

### 3.3. Generation of nAP205-S1 Vaccine Preparations

The IBV S1-SpyTag-SpyTag (0.75 mg) and AP205-SpyCatcher (1.5 mg) were mixed and ligated via the Spy system at 4 °C (Figure 3A). After incubation for 48 h, a new band with an apparent molecular mass of 160 kDa was formed, representing the successful ligation of two proteins (ligation efficiency ~100%) (Figure 3B). Scanning electron microscopy (30× magnification) revealed that the ligated products exhibited a distinct granular morphology (Figure 3C). Subsequently, vaccine solutions were formulated containing conjugated nAP205-S1 and IBV S1-SpyTag-SpyTag at concentrations of 6 μg/0.2 mL, 6 μg/0.2 mL, and 12 μg/0.2 mL, respectively (Figure 3D). Each solution was prepared in a total volume of 24 mL. All samples were subsequently stored at 4 °C for further use.

### 3.4. Immunizations of SPF Chickens with nAP205-S1 Vaccine Candidate

To evaluate the immunogenicity of nAP205-S1, 1-day-old SPF chickens were immunized with nAP205-S1(6 μg/0.2 mL) and IBV S1-SpyTag (6 μg/0.2 mL or 12 μg/0.2 mL), respectively, and then exsanguinated, according to the process listed in Figure 4A. An indirect ELISA assay showed that the anti-IBV antibody response in chicks immunized with nAP205-S1, IBV S1-SpyTag (6 μg/0.2 mL), and IBV S1-SpyTag (12 μg/0.2 mL) turned positive at 7, 10, and 10 dpi, respectively, and the antibody titers in chicks immunized with nAP205-S1 were very similar levels with those in the IBV S1-SpyTag (6 μg/0.2 mL) and IBV S1-SpyTag (12 μg/0.2 mL) groups. The levels of specific antibodies in each immunization group were markedly elevated compared to those observed in the PBS Con group (Figure 4B).

Twenty-eight days post-immunization, the peripheral blood CD4+ T cell and CD8+ T cell counts of chickens in each experimental group were quantified using flow cytometry. The data revealed that the nAP205-S1 immunization group exhibited a significantly higher CD4+ T cell count compared to the two IBV S1-SpyTag immunization groups. In contrast, no significant differences were observed in the CD8+ T cell counts across all experimental groups (Figure 4C). Taken together, these results demonstrate that the immunization of chicks with nAP205-S1 mainly elicits stronger cellular immunity.

### 3.5. Protective Efficacy Against Challenges with Virulent Strains CK/CH/JS/TAHY in Chickens Immunized with nAP205-S1

The protective effectiveness of nAP205-S1 against challenges with a virulent QX-like genotype strain (CK/CH/JS/TAHY) was assessed by infecting six immunized chickens per group at 28 days post-immunization (Figure 4A). Following exposure to approximately ~10^5.5^ EID_50_ of CK/CH/JS/TAHY, no apparent clinical signs were observed in the nAP205-S1-vaccinated group or the IBV S1-SpyTag (6 µg/0.2 mL)-vaccinated group within 8 days post-challenge. In contrast, three chickens (50%) in the control group and one chicken (16.67%) in the IBV S1-SpyTag (12 µg/0.2 mL)-immunized group succumbed to the infection. In addition, none of the unvaccinated chickens kept in the same isolator showed obvious clinical symptoms in the nAP205-S1-immunized group. In contrast, in the PBS control group, two chickens died after the challenge (100%) (Figure 5A). All the dead chickens were autopsied. Hemorrhages (red arrows) were observed in the tracheas (Figure 5B,C) and characteristic “macular nephropathy” lesions were observed in the kidneys (Figure 5B,C). No visible pathological changes were found in the lungs with the naked eye (Figure 5B,C).

At 8 days post-challenge, the surviving chickens in each experimental group were humanely euthanized and subjected to necropsy, which revealed no visible macroscopic lesions in the lungs or kidneys (Figure 5B). With the exception of chickens in the nAP205-S1-immunized group, mild punctate hemorrhages (blue arrows) were observed in the tracheae of some chickens in the PBS control group (3/3), as well as in the IBV S1-SpyTag (6 µg/0.2 mL)-immunized group (3/6) and IBV S1-SpyTag (12 µg/0.2 mL)-immunized group (2/5) (Figure 5B).

The further evaluation of the tissue damage by routine HE sections confirmed that significant differences were observed in the histopathological scores of kidney tissues between the PBS control group and the three immunized groups post-challenge. No significant differences were observed in the histopathological change scores of the tracheal tissues among the experimental groups following a pathogen challenge. The predominant histopathological features included the shedding of mucosal epithelial cells and infiltration of inflammatory cells. The lung tissues of the chickens challenged with the pathogen in each experimental group exhibited minimal histopathological alterations, with no significant differences observed among groups. Notably, mild inflammatory cell infiltration was detected in one out of six chickens in both the PBS control group and the IBV S1-SpyTag (6 µg/0.2 mL) group (Figure 6A). The histopathological change scores in the kidney tissues of chickens from each immunized group were significantly lower than those of chickens in the PBS Con group. In the immunized and challenged groups treated with IBV S1 (6 µg/0.2 mL), IBV S1 (12 µg/0.2 mL), and AP205-S1 (6 µg/0.2 mL) (including two co-housed chickens per group), inflammatory cell infiltration was observed in 1/6, 1/6, and 1/8 of the chickens’ kidneys, respectively. In the PBS Con group (also including two co-housed chickens), inflammatory cell infiltration was detected in 6/8 of the chickens’ kidneys (Figure 6B).

## 4. Discussion

Presently, the primary approach for controlling IBV infection involves vaccine-induced immunity [19]. Commercially available vaccines are either live attenuated or inactivated; however, these vaccines exhibit limited cross-protection. For instance, vaccines based on the Mass genotype are commonly used but provide limited cross-protective immunity against other genotypes, often leading to poor vaccine efficacy and failure [3,5]. Unfortunately, a majority of field isolates, including the QX-like genotype, have the most relevant genotypes currently circulating worldwide and there are no corresponding vaccines. In this study, a genetic engineering technology was applied to successfully construct a novel nanoparticle vaccine with better protection efficiency against the most relevant genotype IBV infection.

Over the past few years, CHO cells have become a preferred system for the expression and preparation of antigen proteins and antibodies due to their ability to perform more complete post-translational modifications [20,21,22], such as glycosylation. This results in recombinant proteins with biological activities that more closely resemble those of native proteins compared to prokaryotic or yeast systems. Coronavirus S glycoprotein is cleaved by host proteases to generate S1 and S2 subunits [23]. The S1 subunit is a highly glycosylated protein [24] and the major inducer of neutralizing antibodies [25]. In this study, the S1 protein was successfully synthesized in CHO cells and secreted into the culture medium. This synthesized protein has excellent immunogenicity. When the immunization dose is as low as 6 μg or 12 μg, it can induce high levels of IgG antibodies and a cell-mediated immune response mainly involving CD4+ cells. Moreover, the protection rate against a challenge with the homologous virulent strain is as high as 83.33% to nearly 100%.

The SpyTag/SpyCatcher protein ligation technology has been applied in the development of various animal and human vaccines [26,27,28]. In this study, the SpyTag/SpyCatche system and AP205-VLPs were employed to develop the nanoparticle antigen nAP205-S1. The AP205 coat protein can not only be efficiently synthesized at a low cost in *E.coli* [29], but also leads to the encapsidation of host cell RNA, thereby exerting an adjuvant effect via TLR7 and TLR8 [30]. Additionally, antigens within the size range of 20 to 200 nanometers exhibit improved drainage to the draining lymph nodes—specialized sites of immune activation—and are more readily taken up by antigen-presenting cells [28]. It might be that when compared with the same immunization dose of IBV S1-SpyTag, nAP205-S1 induced a more significant cellular immune response, mainly involving CD4+ T cells, and demonstrated a superior immune protection effect, achieving a 100% protection rate against the homologous virulent strain.

In this study, we observed that the immunization of chickens with different formats and dosages of vaccine candidates prepared in this study induced high levels of IgG antibodies. Although no gross pathological changes were observed in the nAP205-S1-immunized group, no significant differences were found in the histopathological change scores of tracheal tissues among the three immunized groups. These findings suggest that humoral immunity alone is insufficient to effectively protect against tracheal damage induced by IBV infection; instead, mucosal immunity may play a more critical role in preventing such damage [31,32]. Earlier evidence demonstrated that humoral immunity exerts a pivotal role in the immune protective mechanism against kidney injury. A negative correlation exists between IgG levels and renal injury [31]. This is consistent with our observations that when the S/P ratio of IgG is more than 2.93, no gross pathological lesions are observed in the kidneys. Conversely, when the S/P ratio is less than 1.54, remarkable pathological and histopathologic changes of renal injury may occur. With the aim of improving the protective efficacy against respiratory symptoms and ciliary stasis by leveraging the nanoparticle vaccine nAP205-S1 and the subunit vaccine IBV-S1, novel mucosal immune adjuvants can be explored or the immunization protocols can be optimized. One strategy is to use a simple live vaccine for primary immunization at 1 day old and booster immunization with either the nanoparticle vaccine nAP205-S1 or the subunit vaccine IBV-S1 at 14 days old. This strategy may be useful to address the problems stated above.

## 5. Conclusions

In summary, an efficient antigen expression and assembly system was utilized for the production of the nanoparticle nAP205-S1. Compared to IBV subunit vaccines produced by the baculovirus–insect cell system [33,34] and IBV peptide vaccines [35,36,37], nAP205-S1 vaccines are relatively straightforward to construct and exhibit several advantages, including high safety, high antigen presentation density, and an intrinsic adjuvant effect. Consequently, nAP205-S1 holds considerable promise as a viable alternative to traditional vaccines for the prevention of infectious bronchitis (IB) and other diseases. Given the global diversity of IBV genotypes and the lack of cross-protection among currently available vaccines, it is feasible to simultaneously display IBV antigens of different serotypes or genotypes on the AP205 surface. This strategy highlights a promising avenue for the future development of multivalent IBV vaccines capable of providing broad protection against multiple genotypes of the virus.

## Figures and Tables

**Figure 1 vaccines-13-00802-f001:**
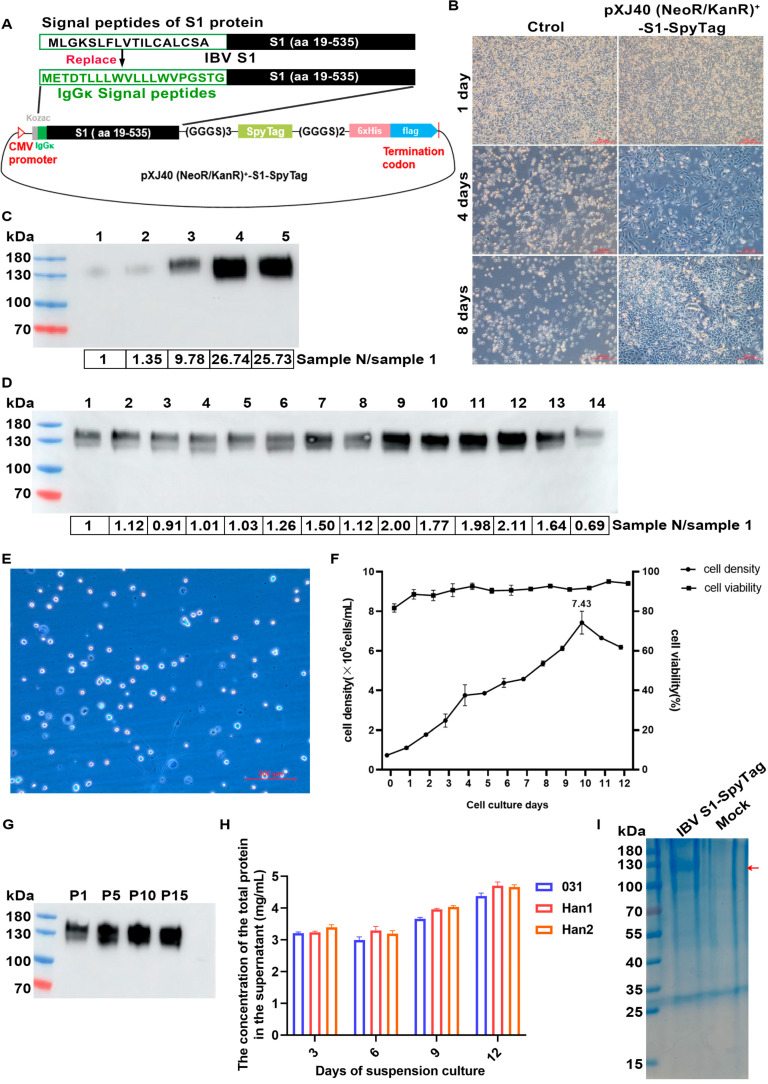
The IBV S1-SpyTag protein was synthesized through serum-free suspension culture technology of CHO cells. (**A**) Schematic diagram of the pXJ40 (NeoR/KanR)+-S1-SpyTag expression plasmid. (**B**) The G418 drug screening approach for positive cells. The cell states of the recombinant plasmid-transfected group and empty vector-transfected group at the 1st, 4th, and 8th days post-transfection. (**C**) Western blot analysis was used to compare the protein expression levels in the supernatants of different CHO-S1 cell clusters during expansion culture using monoclonal antibody anti-IBV S and goat anti-mouse IgG-HRP. (**D**) Western blot analysis was used to compare the protein expression levels in the supernatants of different monoclonal CHO-S1 cell lines during expansion culture using monoclonal antibody anti-IBV S and goat anti-mouse IgG-HRP. (**E**) Diagram illustrating the cell state of CHO stable cell line during serum-free suspension adaptation. (**F**) Monitoring of growth density and viability of serum-free fully suspended CHO stable cells. (**G**) The expression stability of the CHO stable cell line expressing IBV S1-SpyTag was evaluated. Western blot analysis was performed to detect the levels of secreted protein in the supernatant of the CHO stable cell line at passages 1, 5, 10, and 15 during suspension culture, corresponding to p1, p5, p10, and p15, respectively. (**H**) The differences in the expression levels of IBV S1-SpyTag in CHO stable cell lines were compared among three different serum-free media: 031, Han1, and Han2. (**I**) SDS-PAGE analysis of the purity of IBV S1-SpyTag purified protein using 10% gel.

**Figure 2 vaccines-13-00802-f002:**
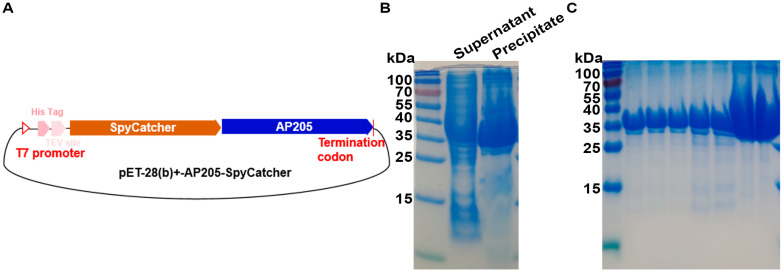
The AP205 coat protein was efficiently synthesized in *E. coli*. (**A**) Schematic diagram of the pET 28(b)+-AP205-SpyCathcer expression plasmid. (**B**) SDS-PAGE analysis of the expression level of AP205-SpyCatcher protein in supernatant and precipitation using 10% gel. The AP205-SpyCatcher construct was transformed into *E. coli* BL21(DE3), which was first grown at 37 °C in LB broth containing kanamycin to OD600 = 0.6~0.8 and then induced with 0.5 mM IPTG for 19 h at 16 °C. (**C**) SDS-PAGE analysis of the purity of IBV S1-SpyTag purified protein using 10% gel. The supernatant containing the synthesized AP205-SpyCatcher protein was filtered through a 0.45 μm filter membrane. The filtrate was then loaded onto a Ni-NTA affinity column and incubated overnight at 4 °C with gentle inversion to ensure efficient binding of the target protein.

**Figure 3 vaccines-13-00802-f003:**
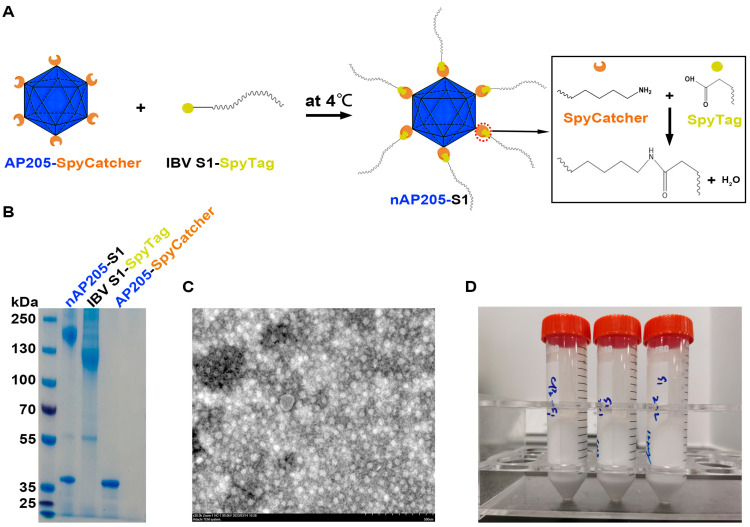
Generation of nAP205-S1 and vaccine preparations. (**A**) Schematic illustration of the in vitro chemical conjugation and assembly of nAP205-S1 nanoparticles mediated by AP205-SpyCatcher and IBV S1-SpyTag. (**B**) SDS-PAGE analysis of the in vitro assembly efficiency of AP205-S1 nanoparticles using 10% gel. nAP205-S1 is a nanoparticle antigen assembled from AP205-SpyCatcher and IBV S1-SpyTag. (**C**) The shape of nAP205-S1 nanoparticles was detected and analyzed by electron microscopy. (**D**) Three vaccines solutions were formulated containing conjugated nAP205-S1 or IBV S1-SpyTag. nAP205-S1 or IBV S1-SpyTag vaccines were at concentrations of 6 μg/0.2 mL, 6 μg/0.2 mL, and 12 μg/0.2 mL, respectively.

**Figure 4 vaccines-13-00802-f004:**
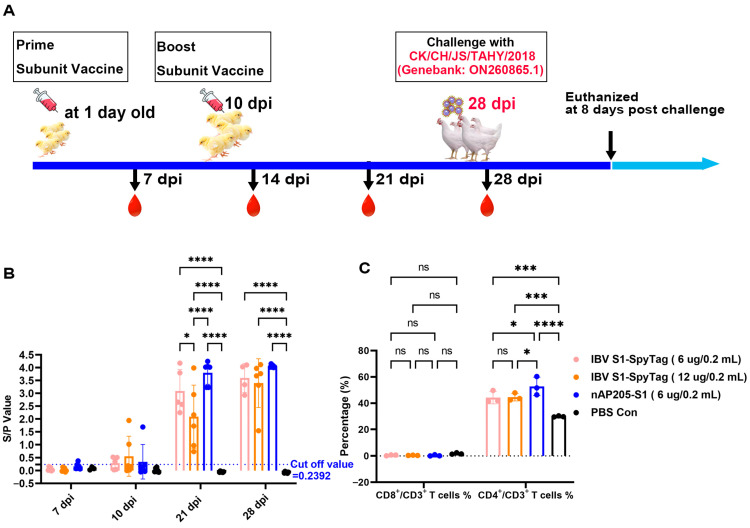
Humoral and cellular immune efficiencies in chickens immunized with nAP205-S1. (**A**) Immunization and exsanguination protocol. Prime-immunization with PBS, IBV S1-SpyTag (6 µg/0.2 mL), IBV S1-SpyTag (12 μg/0.2 mL), and nAP205-S1 (6 μg/0.2 mL) vaccines, respectively. Boost-immunization at 10 days, respectively. Blood samples were collected weekly after immunization. (**B**) Serum antibody titers against infectious bronchitis virus (IBV) were assessed using ELISA at the specified time points. (**C**) The proportions of CD3+, CD4+, and CD8+ T cell populations in peripheral blood samples collected at 28 days post-infection were analyzed by flow cytometry, with statistical evaluation performed using GraphPad Prism 9. The levels of statistical significance are indicated by *p* values, where ns represents non-significance, * *p* < 0.05, *** *p* < 0.001 and **** *p* < 0.0001.

**Figure 5 vaccines-13-00802-f005:**
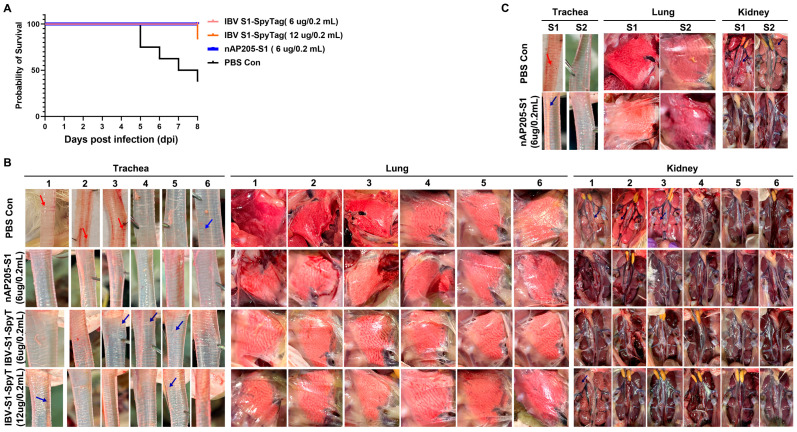
Immune protection efficiency against CK/CH/JS/TAHY. (**A**) Survival curves of six representative viral isolates were generated using 1-day-old SPF chickens. Each group of birds was inoculated via the nasal–ocular route with approximately ~10^5.5^ EID_50_ of one of the six selected isolates. The number of mortalities occurring within 8 days post-challenge was recorded and survival curves were plotted using the Graphpad Prism 9 program. The survival curves of IBV S1-SpyTag (6 µg/0.2 mL), IBV S1-SpyTag (12 µg/0.2 mL), nAP205-S1 (6 µg/0.2 mL), and PBS Con are represented by pink, orange, blue, and black lines, respectively. (**B**) At 8 days post-challenge with CK/CH/JS/TAHY, trachea, lung, and kidney autopsies of dead chickens and all surviving chickens from each experimental group. Red and blue arrows indicate the pathological lesions observed. (**C**) At 8 days post-challenge with CK/CH/JS/TAHY, pathological lesions in the trachea, lung, and kidney autopsies of unvaccinated and unchallenged chickens in the same isolators.

**Figure 6 vaccines-13-00802-f006:**
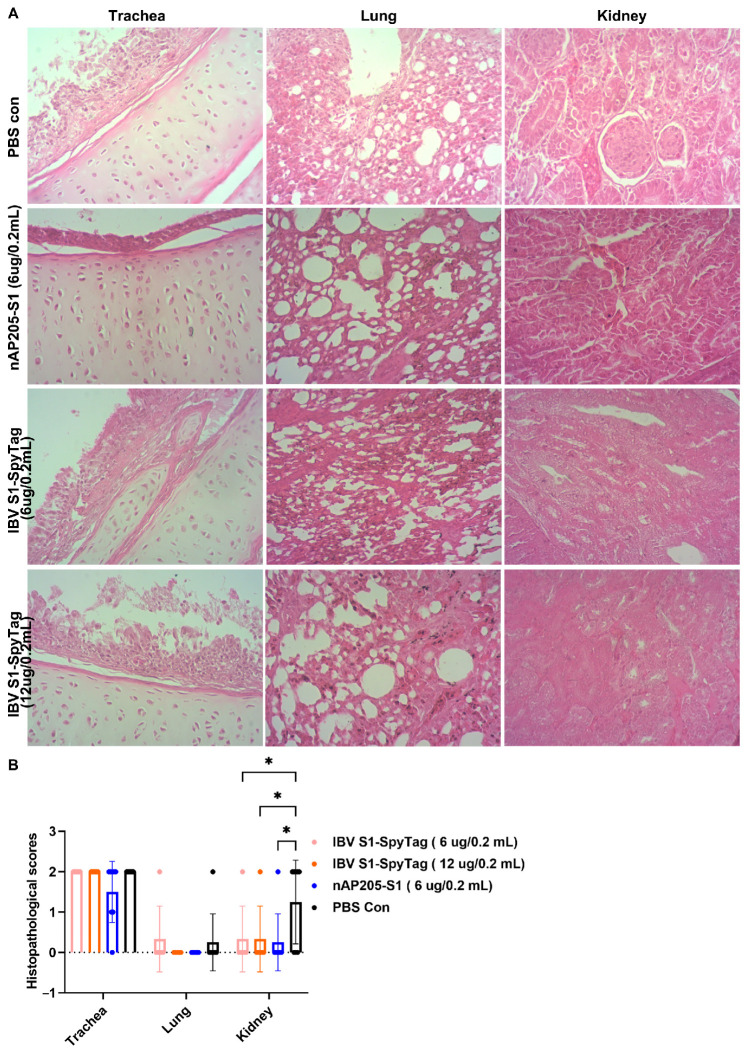
Histopathological analysis of the trachea, lungs, and kidneys was performed on chickens challenged with CK/CH/JS/TAHY. (**A**) Microscopic examination was carried out on tracheal, pulmonary, and renal tissues obtained from deceased chickens or all surviving birds in each experimental group at 8 days post-challenge. (**B**) The pathological tissue change scores of the trachea, lungs, and kidneys in each immune-challenged group were statistically analyzed using Graphpad Prism 9 program at 8 days post-challenge. The levels of statistical significance are indicated by p values, where ns represents non-significance and * *p* < 0.05.

## Data Availability

The original contributions presented in this study are included in this article. Further inquiries can be directed to the corresponding authors.

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
