# Peer review of "Development and Protective Efficacy of a Novel Nanoparticle Vaccine for Gammacoronavirus Avain Infectious Bronchitis Virus"

_vaccines, 2025, doi:10.3390/vaccines13080802_

Round 1

Reviewer 1 Report

Comments and Suggestions for Authors

Comments on Xiong et al Vaccines

General Comments: This is a very well written manuscript describing a very well-designed vaccine and the results of trials. I offer some suggested needed corrections with the major concern being the limitations of the Discussion in discussing the problems yet to solve.

Specific comments:

Line 20 italica E. coli line 119

Abstract – I assume that abbreviations and reagents are defined and explained in detail later in ms.

Line 52 Should adhere to rules pertaining to significant figures - 61.8 or better 62

Line 68&69 define by number in sequence for aspartate and lysine

Line 79 spacing error

Line 148 and following do you need to name the suppliers of the reagents used?

Lines 164 – 168 this is an expensive means of vaccination for commercial use but appropriate for vaccine evaluation

Line 202 Is ***p<0.0001 correct? In Figure 4 you have **** as a level of significance which is not defined

Line 205, 226, 243, 235, etc.  Although the word ‘expressed’ is often used as lab jargon, proteins are synthesized whereas genes are expressed.

Line 214 Southern is a person but western is not, thus lower-case w (except when first word in sentence). The name western blot was proposed by Neal Burnette

Figure 2C what are differences in loading of different lanes?

Figure 5A there should be 4 groups, but I can only see results for two groups. The Figure needs to be redrawn with better clarity. The text on lines 310-314 does not help clarify.

Line 399 I agree that mucosal immunity would be beneficial in conferring complete immune protection. Suggesting administering a live vaccine for initial priming – presumably administered by spray but not so stated - is the only discussed remedy for this deficiency of the constructed vaccine.

Discussion: The discussion is good as far as it goes. The technology is very good and well described. While the vaccine provides a successful means for formulation and inducing after subcutaneous vaccination almost complete protection, there are still three unmet needs. 1) Induction of mucosal immunity - and the authors suggest one reasonable means assuming that mucosal priming with a live attenuated vaccine, presumably administered by a mucosal route -spray – will be boosted by a parenteral subunit vaccine as described in this paper. This can be tested in a subsequent study. 2) Cost of manufacture and administration. This is not mentioned, and I believe would be prohibitively high to enable use of such a vaccine commercially for broiler chickens, although it might be acceptable for layers. This should be a noted problem to solve. In this regard, could the vaccine be administered by in ovo inoculation into 18-day-old chick embryos? This is an established technology and might induce mucosal immunity too. 3) The problem of inducing cross protective immunity to the diverse strains of IBV in circulation. This also needs to be noted as a problem to solve.

References: Quite complete, but there have been other recent attempts to develop vaccines against IBV that may not have been cited and discussed.

Author Response

Comments 1:Line 20 italica E. coli line 119

Response 1:It has been modified to italic as per the suggestion in line 20 and 128

Comments 2:Abstract – I assume that abbreviations and reagents are defined and explained in detail later in ms.

Response 2:It has been modified to italic as per the suggestion.

For example

CHO: Chinese Hamster Ovary in line 109

AP205:capsid protein of Acinetobacter phage AP205 (Gene ID: 956335)in line 124

Comments 3:Line 52 Should adhere to rules pertaining to significant figures - 61.8 or better 62

Response 3:61.77 ± 4.56% has been revised to 61.8 as per the suggestions line 54

Comments 4:Line 68&69 define by number in sequence for aspartate and lysine

Response 4:“an irreversible amide bond is spontaneously formed between the aspartic acid residue on SpyTag and the lysine residue on SpyCatcher”has been revised to“an irreversible amide bond is spontaneously formed between the aspartic acid residue117 on SpyTag and the lysine residue31 on SpyCatcher”

Comments 5:Line 79 spacing error

Response 5:immunizat- ion has been modified to immunization in line 82

Comments 6:Line 148 and following do you need to name the suppliers of the reagents used?

Response 6:had added the suppliers of the reagents

Comments 7:Lines 164 – 168 this is an expensive means of vaccination for commercial use but appropriate for vaccine evaluation

Response 7:This is indeed an expensive approach for commercial vaccination. However, with the optimization of production processes, the cost of manufacturing vaccine antigens will decrease significantly. Additionally, our recent animal studies have demonstrated that an immunization dosage of 3 μg per bird can also achieve excellent immunization efficacy.  

Comments 8:Line 202 Is ***p<0.0001 correct? In Figure 4 you have **** as a level of significance which is not defined

Response 8:***p<0.0001 is correct.and added the sentence as followed:

The specific antibody levels among each immunization group were significantly higher than those of the PBS Con group. In line 288-289

Comments 9:Line 205, 226, 243, 235, etc.  Although the word ‘expressed’ is often used as lab jargon, proteins are synthesized whereas genes are expressed.

Response 9:‘expressed’ has been replaced with 'synthesized' in 212,, etc.

Comments 10:Line 214 Southern is a person but western is not, thus lower-case w (except when first word in sentence). The name western blot was proposed by Neal Burnette

Response 10:Western has been modified to western in line 221

Comments 11:Figure 2C what are differences in loading of different lanes?

Response 11:The different lanes are indicative of the purification efficiencies achieved through elution with varying concentrations of imidazole. It should be noted that the volume of the loaded samples is consistent across all lanes.

Comments 12:Figure 5A there should be 4 groups, but I can only see results for two groups. The Figure needs to be redrawn with better clarity. The text on lines 310-314 does not help clarify.

Response 12:In the nAP205 - S1 immunization group and the IBV S1 - SpyTag (6 μg/0.2 mL) immunization group, no chickens died. As a result, their survival curves were obscured.  .

This description is essential, as the results can reflect the virus excretion patterns of the nAP205 - S1 immunized group and the PBS control group following the challenge.  

Comments 13:Line 399 I agree that mucosal immunity would be beneficial in conferring complete immune protection. Suggesting administering a live vaccine for initial priming – presumably administered by spray but not so stated - is the only discussed remedy for this deficiency of the constructed vaccine.

Response 13:One strategy is to use a simple live vaccine for primary immunization at 1 day old and booster immunization with either the nanoparticle vaccine nAP205-S1 or the subunit vaccine IBV–S1 at 14 days old.(line 411-414)

Comments 14:Discussion: The discussion is good as far as it goes. The technology is very good and well described. While the vaccine provides a successful means for formulation and inducing after subcutaneous vaccination almost complete protection, there are still three unmet needs. 1) Induction of mucosal immunity - and the authors suggest one reasonable means assuming that mucosal priming with a live attenuated vaccine, presumably administered by a mucosal route -spray – will be boosted by a parenteral subunit vaccine as described in this paper. This can be tested in a subsequent study. 2) Cost of manufacture and administration. This is not mentioned, and I believe would be prohibitively high to enable use of such a vaccine commercially for broiler chickens, although it might be acceptable for layers. This should be a noted problem to solve. In this regard, could the vaccine be administered by in ovo inoculation into 18-day-old chick embryos? This is an established technology and might induce mucosal immunity too. 3) The problem of inducing cross protective immunity to the diverse strains of IBV in circulation. This also needs to be noted as a problem to solve.

Response 14:Thank you for your excellent suggestions. We will continue to improve the experiment in the future and have more data in these aspects to support the development of the discussion.

Comments 15:References: Quite complete, but there have been other recent attempts to develop vaccines against IBV that may not have been cited and discussed.

Response 15:Added reference 39:Shao G, Fu J, Pan Y, Gong S, Song C, Chen S, Feng K, Zhang X, Xie Q. Development of a recombinant infectious bronchitis virus vaccine expressing infectious laryngotracheitis virus multiple epitopes. Poult Sci. 2025 Jan;104(1):104578.

Reviewer 2 Report

Comments and Suggestions for Authors

In this manuscript, Xiong et al, addressed the “Development and protective efficacy of a novel nanoparticle vaccine for Gammacoronavirus Avain Infectious Bronchitis Virus”. manuscript is well-structured and relevant to the field of nano vaccine development, especially in the context of Avian Infectious Bronchitis Virus (IBV). The approach uses a SpyTag/SpyCatcher system to assemble antigenic nanoparticles, which is an innovative and well-established method for enhancing immunogenicity. The following comments are provided to help improve the clarity scientific depth, and translational relevance of the manuscript for consideration in MDPI Vaccines.

Reviewer Comments

  1. I suggest authors to characterize formulated nano vaccines for size, zeta potential and long term stability.
  2. On what basis, the authors decided the dose of vaccine immunization. Please clarify
  3. What is the safety concerns of the vaccine formulation?
  4. If possible, I suggest authors to describe brief discussion on cellular immune responses to better understand the overall immunogenicity. The authors assessed CD3, CD4, and CD8. While this provides a general overview of T cell populations, it does not fully capture the functional immune response induced by the vaccine. I suggest authors to add functional such as IFN-γ or IL-2 ELISpot, intracellular cytokine staining, or cytotoxicity assays,
  5. I suggest the authors analyze IgG subtypes to better understand the Th1/Th2 bias of the immune response induced by vaccine formulation

Author Response

Comments 1:I suggest authors to characterize formulated nano vaccines for size, zeta potential and long term stability.

Response 1:Thank you for your suggestion. Due to the lack of relevant instruments and equipment in the laboratory, I will improve the related research work later.

Comments 2:On what basis, the authors decided the dose of vaccine immunization. Please clarify

Response 2:The vaccine dosage was determined based on the findings of our preliminary pre - experiments.

Comments 3:What is the safety concerns of the vaccine formulation?

Response 3:This vaccine is a type of subunit vaccine and exhibits good biosafety. The only immunological side effect is that if the oil adjuvant is not appropriately selected, it may lead to poor absorption, which could potentially affect the quality of chicken meat.  

Comments 4:If possible, I suggest authors to describe brief discussion on cellular immune responses to better understand the overall immunogenicity. The authors assessed CD3, CD4, and CD8. While this provides a general overview of T cell populations, it does not fully capture the functional immune response induced by the vaccine. I suggest authors to add functional such as IFN-γ or IL-2 ELISpot, intracellular cytokine staining, or cytotoxicity assays,

Response 4:We sincerely appreciate your valuable suggestions. Currently, there is a limited availability of domestic ELISpot assay kits for chickens, and their quality is suboptimal, which makes them unable to meet the requirements of our experiments. On the other hand, imported kits are prohibitively expensive, and our research is constrained by budgetary limitations.  

Comments 5:I suggest the authors analyze IgG subtypes to better understand the Th1/Th2 bias of the immune response induced by vaccine formulation

Response 5:We will improve the relevant experiments in our subsequent research based on your suggestions, teacher.